# Uncovering a Novel *cyp*51A Mutation and Antifungal Resistance in *Aspergillus fumigatus* through Culture Collection Screening

**DOI:** 10.3390/jof10020122

**Published:** 2024-02-01

**Authors:** Laís Pontes, Teppei Arai, Caio Augusto Gualtieri Beraquet, Ana Luisa Perini Leme Giordano, Franqueline Reichert-Lima, Edson Aparecido da Luz, Camila Fernanda de Sá, Larissa Ortolan Levy, Cibele Aparecida Tararam, Akira Watanabe, Maria Luiza Moretti, Angélica Zaninelli Schreiber

**Affiliations:** 1School of Medical Sciences, University of Campinas, Campinas 13083-970, São Paulo, Brazil; pontes.laais@gmail.com (L.P.); caioberaquet@gmail.com (C.A.G.B.); analuisa.giordano@gmail.com (A.L.P.L.G.); larilevy@unicamp.br (L.O.L.); ctararam@gmail.com (C.A.T.); moretti.luiza@gmail.com (M.L.M.); 2Division of Clinical Research, Medical Mycology Research Center, Chiba University, Chiba 260-8670, Japan; arai.teppei@chiba-u.jp (T.A.); fewata@faculty.chiba-u.jp (A.W.); 3Department of Medicine, School of Medical Sciences in São José dos Campos—Humanitas, São José dos Campos 12220-061, São Paulo, Brazil; franque.reichert@gmail.com; 4Division of Clinical Pathology, Microbiology Laboratory, University of Campinas Clinical Hospital, Campinas 13083-888, São Paulo, Brazil; edibio@unicamp.br (E.A.d.L.); camilasa@unicamp.br (C.F.d.S.)

**Keywords:** *Aspergillus fumigatus*, azole resistance, Amphotericin B resistance

## Abstract

Background: *Aspergillus fumigatus* is an important concern for immunocompromised individuals, often resulting in severe infections. With the emergence of resistance to azoles, which has been the therapeutic choice for *Aspergillus* infections, monitoring the resistance of these microorganisms becomes important, including the search for mutations in the *cyp*51A gene, which is the gene responsible for the mechanism of action of azoles. We conducted a retrospective analysis covering 478 *A. fumigatus* isolates. Methods: This comprehensive dataset comprised 415 clinical isolates and 63 isolates from hospital environmental sources. For clinical isolates, they were evaluated in two different periods, from 1998 to 2004 and 2014 to 2021; for environmental strains, one strain was isolated in 1998, and 62 isolates were evaluated in 2015. Our primary objectives were to assess the epidemiological antifungal susceptibility profile; trace the evolution of resistance to azoles, Amphotericin B (AMB), and echinocandins; and monitor *cyp*51A mutations in resistant strains. We utilized the broth microdilution assay for susceptibility testing, coupled with *cyp*51A gene sequencing and microsatellite genotyping to evaluate genetic variability among resistant strains. Results: Our findings reveal a progressive increase in Minimum Inhibitory Concentrations (MICs) for azoles and AMB over time. Notably, a discernible trend in *cyp*51A gene mutations emerged in clinical isolates starting in 2014. Moreover, our study marks a significant discovery as we detected, for the first time, an *A. fumigatus* isolate carrying the recently identified TR46/F495I mutation within a sample obtained from a hospital environment. The observed *cyp*51A mutations underscore the ongoing necessity for surveillance, particularly as MICs for various antifungal classes continue to rise. Conclusions: By conducting resistance surveillance within our institution’s culture collection, we successfully identified a novel TR46/F495I mutation in an isolate retrieved from the hospital environment which had been preserved since 1998. Moreover, clinical isolates were found to exhibit TR34/L98H/S297T/F495I mutations. In addition, we observed an increase in MIC patterns for Amphotericin B and azoles, signaling a change in the resistance pattern, emphasizing the urgent need for the development of new antifungal drugs. Our study highlights the importance of continued monitoring and research in understanding the evolving challenges in managing *A. fumigatus* infections.

## 1. Introduction

*Aspergillus* is a genus of saprophytic filamentous fungi with significant medical relevance. Among its various species, *A. fumigatus* stands out as the primary cause of aspergillosis. This disease encompasses a broad clinical spectrum, ranging from allergic syndromes to severe, acute invasive infections. Globally, it affects approximately 8 million individuals annually [1,2].

Azole antifungals are the preferred choice for both the treatment and prophylaxis of *Aspergillus fumigatus* infections [3]. International guidelines specifically recommend Voriconazole (VRC), Itraconazole (ITC), Posaconazole (PSC), and Isavuconazole for addressing aspergillosis [3,4,5]. These antifungals exert their effects by inhibiting the cytochrome P450 enzyme Cyp51, also known as lanosine 14α-demethylase. This enzyme plays a crucial role in converting lanosterol into ergosterol. By reducing ergosterol production, this medication alters the fungal cell membrane’s fluidity, leading to a decrease in fungal cell activity and hindering cell replication [6].

In the past decade, the emergence of azole-resistant *A. fumigatus* has been documented in individuals undergoing prolonged azole treatments and in environmental strains, posing a critical public health concern [2,7,8,9,10,11]. The majority of these resistant strains exhibit mutations in the *cyp51A* gene responsible for the final stage of ergosterol biosynthesis. These resistant strains often carry tandem replications on the promoter region of the *cyp51A* gene, along with non-synonymous point mutations that lead to amino acid changes in the Cyp51A enzyme [12,13,14]. The insertion of 34, 46, or 53 base pairs into the gene’s promoter region, combined with amino acid substitutions, results in varying levels of azole resistance [15,16]. Notably, two commonly observed tandem repeat mutations, TR34 and TR46, are associated with specific point mutations—L98H and Y121/T289A, respectively [13,17]. Patients receiving prolonged azole therapy for *A. fumigatus* infection may experience genetic alterations in the fungus, including tandem repeat and point mutations in the *cyp51A* gene. Moreover, certain amino acid exchanges, such as G54 [18,19], G138 [20], M220 [13,20], and G448 [21], are located near a protein linker, blocking the coupling of most azole antifungal molecules. This interference reduces the interaction between the drug and the fungus, contributing to resistance [22].

Treatment failure represents a significant risk in *A. fumigatus* infection [23]. Guidelines suggest an alternative in the form of Liposomal Amphotericin B (L-AmB), a highly effective antifungal option, although it has a considerable cost. However, resistance to Amphotericin has also been reported [24]. To address cases where azole drugs alone prove ineffective, combinations of antifungals are considered. Echinocandins, including Caspofungin (CAS), Micafungin (MFG), and Anidulafungin (ANI), have demonstrated efficacy in confirmed therapeutic failure, but their use is typically in conjunction with AMB or VRC [3,4,25,26].

In Latin America, the occurrence of resistant *A. fumigatus* has increased, with isolates carrying *cyp51A* mutations detected in Argentina, Colombia, Mexico, Paraguay, and Peru [18,27,28,29,30]. In Brazil, our research team has recently identified strains with *cyp51A* TR34/L98H/S297T/F495I mutations and a G448S substitution [21,31]. After detecting *A. fumigatus* strains displaying resistance attributed to a *cyp51A* mutation within our institution, we conducted resistance surveillance of our *A. fumigatus* collection. This collection, which spans over two decades, had not been previously studied for this specific purpose.

The purpose of this study was to present a retrospective analysis, allowing for the evaluation of the epidemiological profile of clinical and environmental *A. fumigatus* isolates between 1998–2005 and 2014–2021 at our institution. This study aimed to carefully examine the development of resistance to azole antifungals, Amphotericin B, and echinocandins, while also documenting a newly discovered *cyp51A* mutation. Through this investigation, we aim to contribute valuable insights into the evolving landscape of antifungal resistance, uncovering potential challenges and informing future strategies for effective treatment.

## 2. Materials and Methods

### 2.1. Research Institute

The isolates for this study were collected at the University of Campinas Clinical Hospital (HC-UNICAMP) (https://www.hc.unicamp.br/node/76, accessed on 29 January 2024), a tertiary referral hospital in the city of Campinas, and an expansive macro-region of 86 municipalities in the State of São Paulo and neighboring states. Featuring 409 beds and more than 10,000 patients daily, HC-UNICAMP holds a pivotal role as a reference center. This designation involves attending patients with rare diseases and those who are immunocompromised. Given this context, it becomes imperative to actively monitor the resistance patterns of clinical isolates of *A. fumigatus* within our hospital.

### 2.2. Clinical and Environmental A. fumigatus Isolates

For this study, clinical and environmental *A. fumigatus* samples were sourced from the Fungal Investigation Laboratory (LIF) fungal collection at the School of Medical Sciences, University of Campinas. The selection criterion was straightforward: isolates that had not been previously examined in other studies were chosen.

A total of 415 clinical *A. fumigatus* and 63 environmental isolates were collected at HC-UNICAMP in Campinas, São Paulo, Brazil, spanning the years 1998–2005 and 2014–2021 (Figure 1) (Appendix A: *Aspergillus fumigatus* data, including DNA sequencing results and broth microdilution MIC values). Environmental *A. fumigatus* was obtained through air sampling by a modification of the gravity air-settling plate (GASP) method by Krasinski et al. [32,33]. Briefly, conventional 100 × 15 mm disposable Petri dishes containing Sabouraud Dextrose Agar (SDA) were exposed for 60 min in inpatient bedrooms in the bone marrow transplant unit. Weekly sampling occurred while patients were present and receiving care. Petri dishes were then incubated at room temperature and observed for 5 days. Colonies of *A. fumigatus* were selected and purified in SDA.

The isolates were preserved in water at room temperature following the Castellani method [34]. Subsequently, the strains were cultivated on potato dextrose agar (PDA) (Difco Laboratories, Detroit, MI, USA) for a period of 3 to 5 days at a temperature of 35 °C.

### 2.3. Microbiological Morphological Identification

The isolate identification involved an analysis of both macromorphology and micromorphology. Macromorphology was assessed following the growth of SDA and PDA [35].

### 2.4. DNA Extraction and Molecular Identification

Genomic DNA extraction was carried out from isolates cultivated for 48–72 h on SDA plates using a QIAmp^®^ DNA mini-Kit (Qiagen Sciences, Germantown, MD, USA), following the manufacturer’s instructions. For *A. fumigatus* identification, β-tubulin gene amplification was conducted [36,37]. The resulting nucleotide sequences underwent editing using Geneious^®^ 8.1 (Biomatters Ltd., 2015, Newark, NJ, USA) and were utilized as queries in a BLAST search within the NCBI GenBank database (https://blast.ncbi.nlm.nih.gov/Blast.cgi accessed on 29 January 2024).

### 2.5. Antifungal Screening Test for Environmental Strains

To detect resistant isolates, agar plates containing azoles were utilized. Environmental isolates were inoculated onto Mueller–Hinton (MH) agar plates with 2% dextrose. The plates were prepared by incorporating the antifungal agents dissolved in dimethylsulfoxide (DMSO; Sigma-Aldrich, St. Louis, MO, USA) to achieve a final concentration of 2 μg/mL for ITC and VRC and 0.5 μg/mL for POS (Sigma-Aldrich, St. Louis, MO, USA).

The inoculum was prepared following fungal growth for up to 3 days at 35 °C in tubes containing PDA. The final concentration of the inoculum was 2.5 × 10^4^ CFU/mL. In the tests, 20 μL of the inoculum suspension was applied to plates containing the antifungal agents. For growth control, the inoculum was also added to plates without drugs to confirm the viability of the microorganisms. The assessments were conducted after 48 h of incubation at 35 °C.

### 2.6. Broth Microdilution Test (BMD)

The Minimum Inhibitory Concentration (MIC) for azoles and, for echinocandins, Minimum Effective Concentration (MEC) values were determined for all clinical and environmental *A. fumigatus* isolates exhibiting growth in the screening test. The MIC is considered to be when 100% growth inhibition is obtained and the MEC is considered to be when there is a change in the growth pattern following the guidelines outlined in the Clinical and Laboratory Standards Institute M38-A3 [38]. Antifungal susceptibility testing was conducted using pre-prepared dry plates (Eiken Chemical Co., Tokyo, Japan) with inocula of 2.5 × 10^4^ CFU/mL of conidia.

The evaluated antifungal agents were ITC (range 0.015–8 µg/mL), VRC (range 0.015–8 µg/mL), POS (range 0.015–8 µg/mL), Amphotericin B (AMB) (range 0.03–16 µg/mL), MFG (range 0.015–16 µg/mL), and CAS (range 0.015–16 µg/mL). POS, which was not available on the plate, was prepared separately: POS (Sigma-Aldrich, St. Louis, MO, USA) was dissolved in water and then diluted in RPMI 1640 (Sigma-Aldrich).

The MICs for AMB, VRC, ITC, and POS were defined as the lowest concentration causing 100% growth inhibition compared to the drug-free growth control, determined visually after incubation at 35 °C for 48 h. The MECs for MFG and CAS were assessed visually, using the same inoculum concentration after incubation at 35 °C for 24 h. The MEC was considered to be the lowest concentration that led to the growth of small, compact, and rounded hyphae compared to positive growth.

Quality controls, including *Aspergillus flavus* ATCC 204304, *Candida parapsilosis* ATCC 22019, and *Candida krusei* ATCC 6258, were incorporated in each test.

### 2.7. Epidemiological Cutoff Values (ECVs)

The visual method was used to determine the ECV in clinical strains, tracing the distribution of MIC values, and defining the ECV by sight, generally 1 to 2 dilutions beyond the modal MIC value, but before the end point of the distribution, disregarding any tail of the distribution (small amounts of MICs at the top of the distribution), was evaluated as outlined in previous research [39,40].

### 2.8. Detection of cyp51A Mutations

*A. fumigatus* isolates exhibiting MIC >2 µg/mL for ITC and VRC, or POS >1 µg/mL in the BMD, underwent analysis of the *cyp*51A gene. DNA was extracted from 48 h fungal cultures, previously described above, preceding the sequencing of the *cyp*51A gene following established protocols [41,42]. Sequences were aligned with those of an azole-susceptible strain (GenBank accession no. AF338659) using Geneious^®^ 8.1 (2015; Biomatters Ltd., Newark, NJ, USA).

### 2.9. Microsatellite

The study involved the genotypic characterization of resistant strains using nine primer pairs and sequencing approximately 400 base pairs. Repeat numbers of specific regions (2A, 2B, 2C, 3A, 3B, 3C, 4A, 4B, and 4C) were determined from the sequence [43]. A comparison of genotypes was conducted using nine microsatellite markers and a dendrogram was generated through the UPGMA minimum spanning tree algorithm in BioNumerics V7.6 software (Applied Math Inc., Austin, TX, USA).

## 3. Results

### 3.1. Microorganisms

A total of 114 clinical *A. fumigatus* isolates from 1998 to 2005 and 301 isolates from 2014 to 2021 were evaluated. For environmental strains, one *A. fumigatus* isolated in 1998 and 62 isolates isolated in 2015 were evaluated, with detailed data provided in the Appendix A: *Aspergillus fumigatus* clinical isolate number, clinical material, year of isolation, broth microdilution results, and *cyp*51A analysis; Appendix A: *Aspergillus fumigatus* environmental isolate number, year of isolation, and screening test results).

### 3.2. Antifungal Screening Test

Among the 63 environmental *A. fumigatus* isolates assessed in the antifungal screening test, only one isolate (LIF 263E) demonstrated growth across all concentrations of drugs used in this test (VRC, ITC, and POS), as depicted in Figure 2.

### 3.3. Broth Microdilution Test

The MIC and MEC ranges for ITC, VRC, POS, AMB, MFG, and CAS, assessed against 415 clinical isolates, 63 environmental *A. fumigatus* strains, and reference strains, are presented in Table 1.

### 3.4. cyp51A Gene Sequencing

Among the clinical strains, eleven isolates exhibited elevated MICs for at least one azole antifungal evaluated. In contrast, among the environmental isolates, only one (263E) displayed a high MIC for azoles. Subsequently, these isolates underwent analysis of the *cyp*51A gene. The results of the *cyp*51A gene sequencing analysis are shown in Table 2.

### 3.5. Microsatellite

Genotyping analysis was conducted on eleven clinical and one environmental *A. fumigatus* isolates. The phylogenetic tree is shown in Figure 3.

## 4. Discussion

In this study, we analyzed 478 *A. fumigatus* isolates, comprising 415 from clinical sources and 63 from environmental sources. These isolates were obtained from a referral hospital in Campinas, São Paulo, Brazil. The analysis involved confirming the species and antifungal susceptibility, and for resistant isolates, the *cyp51A* gene was sequenced. Additionally, genotyping was performed using microsatellites to assess the genetic variability among the isolates.

We selected 114 clinical *A. fumigatus* from the period 1998 to 2005 and 301 *A. fumigatus* from 2014 to 2021. The results of the BMD for clinical *A. fumigatus*, specifically involving AMB, revealed that 95 (23%) isolates exhibited MIC 2 µg/mL. This finding aligns with a prior study conducted in our institution from 2005 to 2014 [24]. When comparing these results with the EUCAST document [44], which recommends an MIC > 1 µg/mL for AMB to classify *A. fumigatus* as resistant, caution is necessary. This caution stems from the fact that international guidelines recommend the use of AMB as well as the use of other azoles (Isavuconazole and VRC). However, AMB has a high toxicity, in contrast to azoles.

Low MICs were observed for echinocandins in both clinical and environmental strains. Echinocandins are commonly employed in salvage therapy, often as antifungal combination treatments. However, their use as a monotherapy for the initial treatment of invasive aspergillosis is not recommended [25]. To enhance efficacy and reduce the risk of treatment failure, combining an echinocandin with a first-line drug is advised. This combination approach is advantageous as echinocandins not only inhibit hyphal growth but also induce an immunomodulatory effect on the host’s response [45].

In our study, elevated MICs above the ECV were identified for azoles, specifically 35% for ITC, 12% for VRC, and 37% for POS in clinical isolates. According to the EUCAST guidelines [44], MICs > 1 µg/mL for ITC and VRC, and >0.25 µg/mL for POS, are considered indicative of resistance. Additionally, MICs of 2 µg/mL for ITC and VOR and of 0.25 µg/mL for POS decrease into the area of technical uncertainty (ATU).

Our findings align with the EUCAST recommendations for VRC and POS, as the ECVs obtained for these azoles were consistent with EUCAST values indicating resistance. However, for ITC, the ECV was slightly lower at 0.5 µg/mL compared to the EUCAST proposal. Intriguingly, six (1.4%) *A. fumigatus* isolates were observed with MIC values exceeding those proposed by EUCAST.

In the assessment of environmental strains, screening tests using antifungals were employed, which are known for their simplicity and cost-effectiveness [46,47,48]. Among the tested isolates, only one (LIF 263E) demonstrated growth in the screening test. Consequently, this isolate underwent the BMD, revealing resistance to ITC with an MIC > 8 µg/mL, to VRC with an MIC of 4 µg/mL, and to POS with an MIC of 1 µg/mL. Notably, this resistant isolate was collected from the hospital indoors in 1998.

For the analysis of the *cyp51A* gene, isolates were selected based on their MIC values. Those with MICs > 2 µg/mL for ITC and VRC and MICs > 0.5 µg/mL for POS were included. In total, eleven clinical strains and one environmental strain were analyzed.

The mutation identified in the *cyp51A* gene in isolates LIF 3760 and LIF 3763 from the same patient, specifically TR34/L98H/S297T/F495I, was responsible for an increase in MIC values, particularly for ITC. The patient likely acquired this microorganism from the environment [31] because this mutation type is associated with environmental strains [29,49,50,51]. Notably, this specific combination of c*yp*51A mutations has previously been identified in two clinical isolates within our hospital. This finding underscores the presence of resistant *A. fumigatus* carrying *cyp*51A mutations in our facility [31].

For clinical *A. fumigatus* isolates LIF 23A, LIF 472, LIF 479, and LIF 543, in addition to exhibiting high MICs for azoles, an analysis of the *cyp51A* gene did not reveal tandem repeat mutations. Instead, five amino acid changes—F46Y, M172V, N248T, D255E, and E427K—were identified. Snelders and collaborators [13] previously assessed 76 *A. fumigatus* isolates and reported the same set of five amino acid changes. Among the evaluated strains were thirteen isolates with low MIC values for azoles, including one resistant isolate. These amino acid changes have been widely documented in the literature, appearing in both resistant and susceptible isolates [19,24,52]. Importantly, these amino acid changes are not associated with those of the antifungal access target.

Isolate LIF 2328 displayed an MIC > 8 µg/mL for VRC and ITC and an MIC of 1 µg/mL for POS. When analyzing the *cyp51A* gene, no mutations were found, leaving the resistance mechanism still unknown. A study in Japan indicated that 43% of azole-resistant isolates lacked mutations in the *cyp51A* gene [51]. High MIC values for azoles can stem from various mechanisms leading to resistance in *A. fumigatus*, such as efflux pumps, h*ap*E mutation, cholesterol import by *A. fumigatus* [53], and overexpression of the *cyp*51B gene [11]. Recently, it was revealed that a mutation in the mitochondrial ribosome-binding protein Mba1 can confer multidrug resistance to azoles [54]. These non-*cyp51A* mechanisms offer explanations for the existence of resistant *A. fumigatus* strains without mutations in the target gene for azole actions.

By examining clinical isolates resistant to azoles from two distinct periods, 1998 to 2005 and 2014 to 2021, our study reveals a noteworthy trend. Notably, the earlier isolates, despite exhibiting high MICs for azoles, did not display mutations in the *cyp51A* gene. This observation emphasizes a crucial point: the emergence of these mutations occurred later, specifically from 2014 onward. This temporal analysis shows the unique importance of this work, providing a valuable perspective on the evolution of resistance over time.

The environmental *A. fumigatus* isolate (LIF 263E) displaying elevated MIC values for azoles underwent *cyp51A* gene analysis, revealing the presence of TR46/F495I. This amino acid change, specifically the F945I substitution, is known to lead to high MIC values for both azoles and imidazoles, with an MIC > 8 µg/mL [55]. Initially reported in China and Taiwan, this point mutation was later identified in our institution as well [31,55,56]. Interestingly, the year this strain was isolated was 1998. Furthermore, this is the first report of a TR46/F495I combination without the Y121F/A289T mutation. It is still unclear how TR strains occurred. This strain could be the “missing link” between the TR46/Y121F/A289T/F495I strain and the wild-type strain.

In contrast, the pattern observed in the clinical isolates from our hospital differed from that in the environmental mutation. Clinical isolates predominantly exhibited the *cyp51A* TR34/L98H/S297T/F495I mutation. Despite containing the same point mutation, the repetition occurred in gene 34 and involved three additional point mutations.

Despite recent studies, the mechanisms underlying azole resistance development in *A. fumigatus* within the environment remain unclear. The leading hypothesis suggests that resistance may be caused by the indiscriminate use of triazole fungicides [17,57]. Given that *A. fumigatus* is considered a saprophytic fungus, its common exposure to azoles in the environment, whether prolonged or not, can lead to the development of various resistance mechanisms. However, the unexplained predominance of TR34/L98H substitution suggests that selective pressures in the environment or the specific properties of these isolates may contribute to their extended survival [17].

This marks the initial discovery of the 46 tandem repeat combined with an F495I point mutation within the indoor hospital environment. The strain, isolated in 1998, affirms the presence of *A. fumigatus* resistance to azoles in our country since the late 1990s. Despite its environmental origin, this microorganism was isolated from a patient’s room, showing the potential for transmission within hospital settings.

A comparison of STR patterns found in resistant isolates was undertaken in this study; it was possible to observe the formation of seven different clusters. However, some grouped *A. fumigatus* isolates suggested that they have the same clonal origin. (1) Clusters LIF 3760 and LIF 3763 and (2) clusters LIF 23A, LIF 543, and LIF 479 have the same pattern of microsatellites. Additionally, they have the same mutations in *cyp51A*. The presence of a clonal complex indicates that *A. fumigatus* has the same clonal origin and that there may be differences of 1 to 2 STs in the genome. The results obtained in this work corroborate previous studies that demonstrated high disparity in *A. fumigatus* populations between patients and diversity within the same patient [58,59,60]. Some isolates outside the clonal complex present more than 2 STs of difference from one to the other; however, this can be elucidated by the fact that the STRAf 3A and STRAf 3C markers are unstable, which is an important factor that must be evaluated with caution when interpreting the results [61].

The rapid emergence of azole-resistant strains of *A. fumigatus* is a growing concern. Resistance mechanisms are crucial for microbial survival. We believe in immediate and comprehensive studies to enhance our comprehension of the development and transmission of azole-resistant *A. fumigatus*. National studies are imperative, given the absence of data from other reference centers. Environmental and clinical resistance surveillance for *A. fumigatus* is vital.

This study investigated the epidemiological profile of *A. fumigatus* clinical isolates in a Brazilian hospital, covering two time periods (1998–2005 and 2014–2021). Among the 415 isolates analyzed, two exhibited the TR34/L98H/S297T/F495I mutation in the *cyp*51A gene, and, notably, this study identified the first environmental strain of *A. fumigatus* carrying the TR46/F495I mutation within the hospital. The presence of this mutation pattern has significant implications for clinical management, especially if infections with such mutations are reported. Despite a low prevalence of *cyp*51A mutations in our center, the observed high MIC values for azoles may be attributed to other resistance mechanisms, such as efflux pumps and mutations in the *cyp*51B gene or in the *hmg*1 gene. The identification of elevated MIC values for both AMB and azoles suggests a changing resistance pattern in our institution, emphasizing the urgency of discovering new antifungal drugs and resistance mechanisms. Continued surveillance of new resistant isolates is deemed necessary.

## Figures and Tables

**Figure 1 jof-10-00122-f001:**
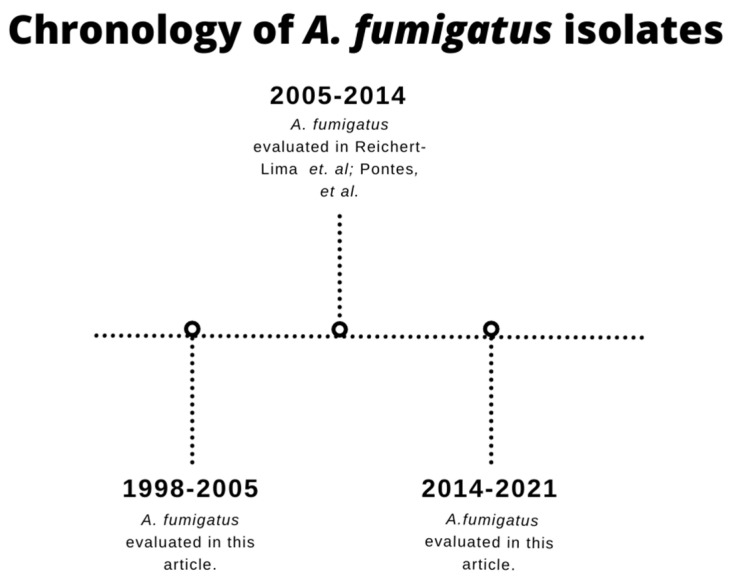
The timeline of *A. fumigatus* was evaluated in this study and other articles by our research group [24,31].

**Figure 2 jof-10-00122-f002:**
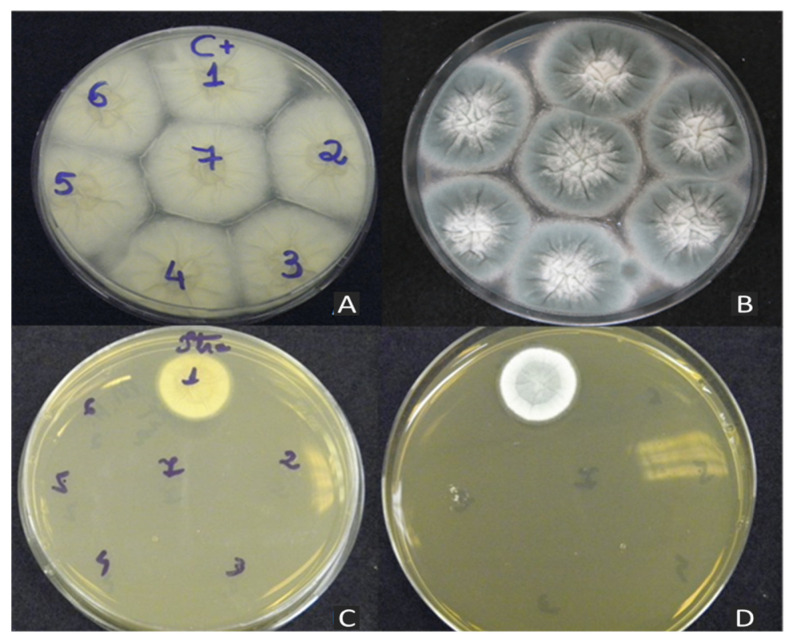
Screening test with Mueller–Hinton agar conducted in 90 × 15 mm Petri dishes. The numbers shown on the plate correspond to the environmental isolates tested. (**A**,**B**) Positive control test without antifungals: (**A**) back of plate; (**B**) front of plate. (**C**,**D**) Screening test with ITC at a concentration of 2 μg/mL, highlighting the presence of the resistant isolate 263E: (**C**) back of plate; (**D**) front of plate.

**Figure 3 jof-10-00122-f003:**
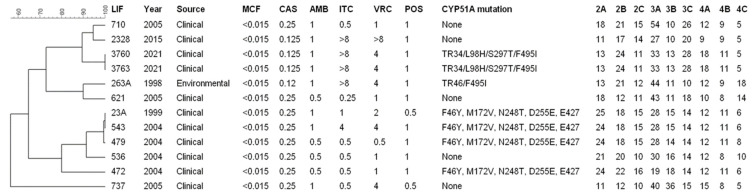
Genetic correlation between the azole-resistant isolates found in this study. Phylogenetic tree prepared with unweighted pair group method with arithmetic mean (UPGMA) clustering using Phyloviz 2.0a. Short tandem repeat markers 2A, 2B, 2C, 3A, 3B, 3C, 4A, 4B, and 4C.

**Table 1 jof-10-00122-t001:** Distribution of MICs and MECs (µg/mL) for all tested antifungal agents against all clinical isolates and one resistant environmental *A. fumigatus* strain.

*Aspergillus fumigatus*	Antifungal Agent	MIC/MEC Distribution (µg/mL)							
		≤0.015	0.125	0.25	0.5	1	2	4	>8
**Clinical (415)**	**ITC**			39	231 *	139	2	1	3
	**VRC**			2	104	259 *	44	5	1
	**POS**		27	233 *	146	9			
	**AMB**			5	55	260 *	95		
	**CAS**		15	397 *	3				
	**MFG**	415 *							
**Environmental (263E)**	**ITC**								1
	**VRC**							1	
	**POS**					1			
	**AMB**					1			
	**CAS**		1						
	**MFG**	1							
**Reference Strains**	**MIC/MEC Distribution (µg/mL)**								
	**ITC**	**VRC**	**POS**	**AMB**		**CAS**		**MFG**	
*Candida parapsilosis* ATCC 22019	0.125	0.25	0.06	1		1		0.25	
*Candida krusei* ATCC 6258	0.125	0.5	0.06	1		1		0.06	
*Aspergillus flavus* ATCC 204304	0.5	0.25	0.25	1		0.25		≤0.015	

MIC: Minimum Inhibitory Concentration for azoles and AMB; MEC: Minimum Effective Concentration for Equinocandins; * Epidemiological Cutoff Values (ECVs) for each antifungal agent in this sampling: AMB: Amphotericin B; ITC: Itraconazole; VRC: Voriconazole; POS: Posaconazole. MEC: Minimum Effective Concentration; MFG: micafungin; CAS: caspofungin. Breaking point for resistance > 2 µg/mL for ITC and VRC; POS > 0.5 µg/mL; AMB > 1 µg/mL; CAS and MFC = not available [39,44].

**Table 2 jof-10-00122-t002:** Antifungal susceptibilities of azole-resistant *A. fumigatus* isolates and *cyp*51A gene analysis.

LIF	Year	Mutations in *cyp*51A	MEC/MIC (µg/mL) *					
			MCF	CAS	AMB	ITC	VRC	POS
23A	1999	F46Y, M172V, N248T, D255E, E427	≤0.015	0.25	1	0.5	4	0.5
472	2004	F46Y, M172V, N248T, D255E, E427K	≤0.015	0.25	0.5	0.5	1	1
479	2004	F46Y, M172V, N248T, D255E, E427K	≤0.015	0.25	0.5	0.5	0.5	1
536	2004	None	≤0.015	0.25	0.5	0.5	1	1
543	2004	F46Y, M172V, N248T, D255E, E427K	≤0.015	0.25	1	4	4	1
621	2005	None	≤0.015	0.25	0.5	0.25	1	1
710	2005	None	≤0.015	0.25	1	0.5	1	1
737	2005	None	≤0.015	0.12	0.5	1	4	0.5
2328	2015	None	≤0.015	0.125	1	>8	>8	1
3760	2021	TR34/L98H/S297T/F495I	≤0.015	0.125	1	>8	4	1
3763	2021	TR34/L98H/S297T/F495I	≤0.015	0.125	1	>8	4	1
**Environmental**								
263E	1998	TR46/F495I	≤0.015	0.12	1	>8	4	1

* MIC: Minimum Inhibitory Concentration for azoles and AMB; MEC: Minimum Effective Concentration for Equinocandins; AMB: Amphotericin B; ITC: Itraconazole; VRC: Voriconazole; POS: Posaconazole. MEC: Minimum Effective Concentration; MFG: Micafungin; CAS: Caspofungin.

## Data Availability

Data are contained within the article.

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
