# Peer review of "Uncovering a Novel cyp51A Mutation and Antifungal Resistance in Aspergillus fumigatus through Culture Collection Screening"

_jof, 2024, doi:10.3390/jof10020122_

Round 1

Reviewer 1 Report

Comments and Suggestions for Authors

In this study, a total of 478 A. fumigatus isolates, including clinical and environmental strains, form a hospital in Campinas, Brazil were analyzed, in order to determine their antifungal resistance, and the presence of mutations in the cyp51A gene in resistant strains. 

In general, the introduction is well presented, with al the information needed to understand the background of the paper's objective; most of the methods are well explained and the experimental approach was properly designed; the results are satisfactory analyzed and explain the progressive increased in MIC values for azoles and AMB over time in these strains. Also, the authors present the first report of an A. fumigatus strain isolated from the hospital with the TR46/F495I mutation in the cyp51A gene; and the discussion is consistent with the results, explaining them correctly and comparing them with the results of previous reports. In conclusion, the data presented in this paper highlights the importance of monitoring the A. fumigatus resistant strains and the development of new treatments for this fungal pathogen.

However, I ask the authors to address the following comments and questions:

Abstract

Line 21: I think an “and” is missing, shouldn’t it say, “hospital and environmental sources”?

Line 28: what is a thorough analysis?

Line 35: change the word “signals” for “shows”

Line 36: change the word “imperative” for “need”

Introduction

Line 47: change the word “treating” for “treatment”

Line 56: remove the "," and change “samples” for “strains”

Line 59: change “of” for “on”                               

Line 68: remove the ","

Line 84: what kind of assessment? Genetic?

Line 86: remove the "s" from the word “presents”

Line 90: remove the word “focused”

Materials and methods

Line: 99: change the word “for” for “in” and put a "," after Campinas 

Line 102: remove the word “to”

Line 107: add a "," after “(LIF)” and change “fungi” for “fungal”

Line 113: add a dot after “values)”

line 128: and how was the micromorphology analysis performed?

Line 137: the antifungal screening test for clinical strains was not performed?

Line 151: add “and environmental A. fumigatus isolates” after clinical, so it reads like this: “for all clinical and environmental A. fumigatus isolates”

Line 158: what do you mean with POS was not available on the plate? Why wasn’t available?

Line 167 and 68: why were yeast cells, such as Candida, included as controls? Why wasn’t the reference strain of A. fumigatus included in the controls? If the strains that you are working with belong to this specie

Line 169: it would be nice if you could explain briefly the epidemiological cutoff values methodology

Line 173: add “and” between ITC and VRC

Results

Line 187: what about the clinical isolates? Only the environmental isolates are mentioned in this section of microorganisms

Line 189: add “were used/were analyzed/were included in this study” after 2021

Line 190: A. fumigatus is not in italic

Line 196: there was growth in the presence of all concentrations of all drugs, or only for ITC at a concentration of 2ug/mL? as shown in the description of figure 2

Line 194: why wasn’t the antifungal screening test performed for the clinical strains?

Line 206: remove “s follows” and add “strain”

In table 1, the environmental strain is the isolate 263E? it would be good to mention it on the table. Also, what happened with the other 62 environmental strains? do they have MIC and MEC values lower than ≤0.015?

Finally, it would be nice to include on the tables or to mention on the text, what are the reference values being used to determine resistance. I know is mentioned in the discussion (line 248), but when you are looking at the tables, is not clear how you are determining resistance.

Discussion

Line 266: the two isolates mentioned here were isolated from the same patient?

Supplementary material

Table S1: the names of the drugs are different in the first line of the table and the line 418. Why are there some strains where the CYP51A sequence was not evaluated?

Table S1 and S2: It is not clear which strains are presented in which supplementary tables. It would be nice if you could identify the environmental and clinical strains, or at least mention what strains are being presented in each table

Comments on the Quality of English Language

There are a few orthographic errors mentioned above that need to be corrected, but in general, the text is well written and easy to understand. 

The errors are marked in the PDF file attached as well 

Author Response

Dear Editorial Office,

We hereby resubmit the manuscript, ID jof-2815708, entitled “Uncovering a Novel cyp51A Mutation and Antifungal Resistance in Aspergillus fumigatus through Culture Collection Screening”, by Laís Pontes, Teppei Arai, Caio Augusto Gualtieri Beraquet, Ana Luisa Perini Leme Giordano, Franqueline Reichert-Lima, Edson Aparecido da Luz, Camila Fernanda de Sá, Larissa Ortolan Levy, Cibele Aparecida Tararam, Akira Watanabe, Maria Luiza Moretti, and myself, for further consideration.

We begin by thanking the Editorial Office and reviewers for their time and honest evaluation of the original version of this manuscript. The comments and suggestions have greatly improved the clarity and quality of our work. Therefore, we believe that the revised version meets the standards of the Journal of Fungi.

Reviewer 1

In this study, a total of 478 A. fumigatus isolates, including clinical and environmental strains, form a hospital in Campinas, Brazil were analyzed, in order to determine their antifungal resistance, and the presence of mutations in the cyp51A gene in resistant strains. 

In general, the introduction is well presented, with al the information needed to understand the background of the paper's objective; most of the methods are well explained and the experimental approach was properly designed; the results are satisfactory analyzed and explain the progressive increased in MIC values for azoles and AMB over time in these strains. Also, the authors present the first report of an A. fumigatus strain isolated from the hospital with the TR46/F495I mutation in the cyp51A gene; and the discussion is consistent with the results, explaining them correctly and comparing them with the results of previous reports. In conclusion, the data presented in this paper highlights the importance of monitoring the A. fumigatus resistant strains and the development of new treatments for this fungal pathogen.

However, I ask the authors to address the following comments and questions:

Authors: We thank Reviewer 1 for the comments and suggestions for improvements to the abstract.

Abstract

Line 21: I think an “and” is missing, shouldn’t it say, “hospital and environmental sources”?

Authors: The referee's observation about the potential ambiguity in the sentence is valid. However, in this case, the omission of "and" is intentional. The goal of the sentence is to highlight that the isolates were obtained from hospital environments, from bone marrow transplant patient rooms. Adding "and" might dilute this emphasis and imply a broader scope that includes other environmental sources. Therefore, the sentence structure has been chosen to distinguish the strains obtained from hospital environments and patient samples.

Line 28: what is a thorough analysis?

Authors: In this sentence, the word was originally employed to convey that we conducted a comprehensive and detailed analysis. However, to prevent potential misunderstandings, this phrase has been excluded. 

Line 35: change the word “signals” for “shows”

Line 36: change the word “imperative” for “need”

Authors: We thank the referee for suggesting improvements to the abstract. We have changed the conclusion and removed these words.

Introduction

Line 47: change the word “treating” for “treatment”

Line 56: remove the "," and change “samples” for “strains”

Line 59: change “of” for “on”                               

Line 68: remove the ","

Line 84: what kind of assessment? Genetic?

Line 86: remove the "s" from the word “presents”

Line 90: remove the word “focused”

Authors: Done.

Materials and methods

Line: 99: change the word “for” for “in” and put a "," after Campinas 

Line 102: remove the word “to”

Line 107: add a "," after “(LIF)” and change “fungi” for “fungal”

Line 113: add a dot after “values)”

Authors: Done

line 128: and how was the micromorphology analysis performed?

Authors: The micromorphology analysis was conducted by examining the detailed characteristics of each selected A. fumigatus for this study, including features such as conidiophore, conidia, phialides, etc. This analysis was deemed necessary due to the presence of numerous fungi that resemble A. fumigatus when observed at a macroscopic level. Examining these finer micromorphological details allowed for a more precise and accurate identification of Aspergillus fumigatus, distinguishing it from other fungi that might share similar macroscopic traits. 

Line 137: the antifungal screening test for clinical strains was not performed?

Authors: No, the antifungal screening test for clinical strains was not conducted in our study. We focused solely on screening environmental strains. Clinical strains, on the other hand, underwent testing using the gold standard Broth Microdilution method as part of our experimental procedures. 

Line 151: add “and environmental A. fumigatus isolates” after clinical, so it reads like this: “for all clinical and environmental A. fumigatus isolates”

Authors: Done.

Line 158: what do you mean with POS was not available on the plate? Why wasn’t available?

Authors: When mentioning that "POS was not available on the plate," it indicates that the Dry plates from Eiken-Japan, the commercial brand, did not contain posaconazole (POS) for susceptibility testing. The commercial test comes with lyophilized antifungals (Itraconazole, Voriconazole, Amphotericin B, Micafungin, Fluconazole, Caspofungin, and 5-FC) and the testing process only involved adding the culture medium and inoculum. Since the brand does not include POS in its pre-prepared plates, we conducted an in-house broth microdilution assay specifically for this drug.

Line 167 and 168: why were yeast cells, such as Candida, included as controls? Why wasn’t the reference strain of A. fumigatus included in the controls? If the strains that you are working with belong to this specie

Authors: Yeast cells, including Candida species, were included as controls in line with the CLSI M38-A3 document recommendations. This document advises the use of Aspergillus flavus ATCC, Candida parapsilosis ATCC, and Candida krusei ATCC as quality control strains for testing. As our study aligns with the guidelines outlined in this document, we opted for the standard strains it suggests. Therefore, the reference strain of Aspergillus fumigatus was not included in the controls because we followed the specific recommendations of the CLSI guidelines.

Line 169: it would be nice if you could explain briefly the epidemiological cutoff values methodology

Authors: We have added the appointed information in the revised version of the document. 

Line 173: add “and” between ITC and VRC
Authors: Done.

Results

Line 187: what about the clinical isolates? Only the environmental isolates are mentioned in this section of microorganisms

Authors: We have added this information.

Line 189: add “were used/were analyzed/were included in this study” after 2021

Line 190: A. fumigatus is not in italic

Authors: Done.

Line 196: there was growth in the presence of all concentrations of all drugs, or only for ITC at a concentration of 2ug/mL? as shown in the description of figure 2

Authors: The growth observed in Figure 2, specifically for Itraconazole (ITC) at a concentration of 2 ug/mL, is only depicted for illustrative purposes. In reality, there was growth in the presence of all concentrations for all the antifungals evaluated in the screening test. These antifungals include Itraconazole, Posaconazole, and Voriconazole.

Line 194: why wasn’t the antifungal screening test performed for the clinical strains?

Authors: We chose not to conduct the antifungal screening test for the clinical strains due to their clinical origin. Instead, we directly employed the Microdilution in Broth method for these strains, which is considered the gold standard method in the clinical setting.

Line 206: remove “s follows” and add “strain”

Authors: Done.

In table 1, the environmental strain is the isolate 263E? it would be good to mention it on the table. Also, what happened with the other 62 environmental strains? do they have MIC and MEC values lower than ≤0.015?

Authors: Yes, the environmental strain referred to in the table is indeed the isolate 263E. However, as the broth microdilution was conducted only for the isolate that exhibited growth in the screening test, we did not determine the exact MIC values for the other 62 environmental strains for the other drugs. Consequently, we have included the MIC value only for isolate 263e. 

Finally, it would be nice to include on the tables or to mention on the text, what are the reference values being used to determine resistance. I know is mentioned in the discussion (line 248), but when you are looking at the tables, is not clear how you are determining resistance.

Authors: We have addressed the concern regarding the determination of resistance values. The relevant information about the reference values used to determine resistance is now included in the table legends for clarity. 

Discussion

Line 266: the two isolates mentioned here were isolated from the same patient?

 Authors: Yes, the two isolates mentioned here were isolated from the same patient. This information has been included in the text for clarification. 

Supplementary material

Table S1: the names of the drugs are different in the first line of the table and the line 418. Why are there some strains where the CYP51A sequence was not evaluated?

Authors: We have updated the names of the antifungals in the supplementary table for consistency. Regarding the evaluation of the cyp51A sequence, it was selectively performed only for isolates that exhibited high MIC values for azoles in the broth microdilution test. Since resistance is predominantly associated with mutations in the cyp51A gene (approximately 98% of cases), we strategically decided not to sequence all isolates in the study. Instead, our focus was on sequencing those isolates that demonstrated resistance to obtaining relevant genetic information.

Table S1 and S2: It is not clear which strains are presented in which supplementary tables. It would be nice if you could identify the environmental and clinical strains, or at least mention what strains are being presented in each table.

Authors: We have addressed the concern about clarity in the presentation of strains in the supplementary tables. The supplementary material now includes information that identifies whether the strains presented are environmental or clinical.

The revised manuscript reviewed can be found in the attachment. 

Reviewer 2 Report

Comments and Suggestions for Authors

please see file enclosed

Author Response

Dear Editorial Office,

We hereby resubmit the manuscript, ID jof-2815708, entitled “Uncovering a Novel cyp51A Mutation and Antifungal Resistance in Aspergillus fumigatus through Culture Collection Screening”, by Laís Pontes, Teppei Arai, Caio Augusto Gualtieri Beraquet, Ana Luisa Perini Leme Giordano, Franqueline Reichert-Lima, Edson Aparecido da Luz, Camila Fernanda de Sá, Larissa Ortolan Levy, Cibele Aparecida Tararam, Akira Watanabe, Maria Luiza Moretti, and myself, for further consideration.

We begin by thanking the Editorial Office and reviewers for their time and honest evaluation of the original version of this manuscript. The comments and suggestions have greatly improved the clarity and quality of our work. Therefore, we believe that the revised version meets the standards of the Journal of Fungi.

Reviewer 2

The authors report on type and the distribution of Aspergillus resistance mutations including clinical and environmental isolates. The manuscript is mainly well written and provides potentially important new data on this important issue.

Authors: We thank you for your thoughtful review of our manuscript. Your positive feedback on the overall quality and recognition of the potentially significant new data is appreciated. Your constructive comments provide valuable insights that will be carefully considered in refining our work. We are committed to addressing your suggestions to ensure the final version aligns with the high standards of the Journal of Fungi.

Do the authors have clinical data on the patients from whom the isolates were identified?

Authors: Unfortunately, we do not possess clinical data for the isolates identified between 1998 and 2004, as our system transitioned to automation in 2010. Regrettably, medical records for patients before 2010 are unavailable. Consequently, the clinical data we have is limited to patients included after 2014. We have opted not to incorporate this data into the manuscript to avoid creating a gap in the information related to earlier patients. We appreciate your understanding and are open to any further guidance you may have regarding this limitation.

Are there any information on the local distribution of pathogens including the development in the period studied?

Authors: Information on the local distribution of pathogens, including their development over the studied period, is available in the supplementary material. We have included comprehensive clinical material data in supplementary Table S1, which provides a detailed account of the results for all the microorganisms evaluated in this manuscript. We trust that this supplementary information will address your query adequately. Please feel free to reach out if you have any further questions or require additional clarification.

What is the authors hypothesis that resistance mutations mainly emerged after the year 2014?

Authors: We appreciate your question regarding the hypothesis behind the emergence of resistance mutations primarily after the year 2014. Our hypothesis is rooted in the potential impact of several factors. First, we posit that the indiscriminate use of pesticides in nature since 2014 may be a contributing factor. This widespread use has been associated globally with resistance to azole antifungals. Additionally, the prophylactic use of azoles in immunocompromised patients is another significant factor. The substantial selective pressure exerted on A. fumigatus strains, both in natural environments and within human hosts, may be fostering the development of resistance and mutations in cyp51A. 

Are there more information on occurrence of cyp p51 mutation in other regions of the world?

Authors: Certainly, there are reports of the occurrence of cyp51A mutations in various regions worldwide. In the introduction of our manuscript, we have cited several instances of mutations in cyp51A from different parts of the world. However, it's noteworthy that the new mutation discussed in our manuscript, namely TR46/F495I, has not been previously reported. 

In which clinical situations might testing for these mutations be helpful?

Authors: We propose that testing for mutations in the cyp51A gene should be considered in clinical situations where patients do not respond to treatment. International guidelines recommend monitoring for Aspergillus fumigatus infections using markers like galactomannan, along with diagnostic methods such as sputum culture and bronchoalveolar lavage. From these cultures, Aspergillus fumigatus strains can be isolated, enabling antifungal susceptibility testing. Subsequent sequencing of the cyp51A gene is recommended in cases where resistance is suspected. 

Figure 3: The meaning of 2a-4c of the Bioanalyzer program should be explained for the reader not familiar with this program.

Authors: Thank you for highlighting the need for clarification in Figure 3 regarding the Bioanalyzer program. We have revised the figure legend to include information about the meaning of 2a-4c in the Bioanalyzer program. 

Page 8: “This carefulness stems from the fact that AMB is less commonly used due to its toxicity and is typically reserved as a second-line treatment when first-line options such as azoles are ineffective.“ International guidelines recommend both L-Amb and isavuconazole/voriconazole with a grade-A recommendations.

Authors: Thank you for bringing attention to the sentence in question. We have duly reformulated the statement to align with international guidelines. 

Page 8: “The patient likely acquired this microorganism from the environment [31], because this mutation type is associated with environmental strains”. I assume that the majority of patients with invasive Aspergillosis acquire the pathogens from the environment, as also shown by the fact that construction works increase the incidence of IFD.

Authors: Thank you for highlighting this crucial aspect. Indeed, your assumption is well-founded. Mutations featuring tandem repeats in their composition are presumed to originate from the environment, contrasting with amino acid changes like G448S, M220, etc., which are directly associated with prolonged in vivo azole use. Notably, prior studies have substantiated that patients on continuous Voriconazole usage can develop A. fumigatus strains harboring the G448S mutation in the cyp51A gene. 

Table 2. Have the authors more information on the significance of the distinct amino acid changes (cyp51A substitution) regarding MIC, resistances patterns and clinical associations?

Authors: Unfortunately, we do not have additional information beyond what is presented in this manuscript regarding the significance of the distinct amino acid changes in cyp51A concerning MIC, resistance patterns, and clinical associations. Our research group has plans to delve into these aspects in the future, and we are committed to expanding our understanding in subsequent studies. 

Do the authors have more data on the use of novel antifungals such as fosmanogepix, ibrexafungerp, rezafungin or isavuconazole in patients with Aspergillus resistance mutations?

Authors: We currently do not possess data on the use of novel antifungals such as fosmanogepix, ibrexafungerp, rezafungin, or isavuconazole specifically in patients with Aspergillus resistance mutations. Notably, these new drugs have not been tested for their efficacy against resistant strains of Aspergillus fumigatus. In terms of clinical use, the Brazilian Unified Health System (SUS) has recently implemented Isavuconazole for the treatment of invasive Aspergillosis. However, we anticipate that with the initiation of its use in our country, resistance to this antifungal may emerge. Consequently, we are preparing to commence testing the new drugs available in our region.

The revised manuscript can be found in the attachment.

Reviewer 3 Report

Comments and Suggestions for Authors

The manuscript of Laís Pontes and co-authors is mainly a study of a collection of A. fumigatus strains. The authors analyzed the number of resistant strains and strains carrying mutations in the cyp51A gene. But it is still unclear from the work what fundamentally new data and conclusions were obtained in this work. The spread of azole-resistant fungal strains is widely known. And it is known that there are different mechanisms for the development of resistance. It is not clear why the authors chose to study mutations in cyp51A gene. The authors themselves write that mutations in this gene are not so common in their collection. And what conclusions can be drawn from the work done.

The main comments

1)              The title of the manuscript is too long.

2)              Abstract.

-It is not clear from the abstract what cyp51A gene is and what it is responsible for. About mutation: TR46/F495I requires clarification (tandem repeat of 46 bp?).

-It is not clear from the conclusion what new has been done in this work. The spread of resistant strains of fungi is widely known.

3) Materials and methods section.

- It’s still not entirely clear what MEC is and what it shows.

- “Quality controls, including Aspergillus flavus ATCC 204304, Candida parapsilosis ATCC 167 22019, and Candida krusei ATCC 6258 were incorporated in each test.” Why was not the collection strain Aspergillus fumigatus used for this purpose?

4) Results section.

- There is no description of the actual results obtained. Some information should be transferred from the Discussion section or the Results and Discussion sections should be combined.

- Figure 2. Replace A1, A2, B1, B2 with A, B, C, D. What exactly is on the petri dishes? What do the numbers on the petri dishes correspond to? What is the difference between A1 and A2 (B1 and B2)?

- Table 1. Сheck the table name. What values ​​are given in the table – MICs or MECs? What is “MIC/MEC Distribution (μg/mL)”?

- There are no table S1,S2 legends.

- Сhange the name “3.4. cyp51A Analysis”. Аnalysis of what?

- Table 2. Сhange table name “In vitro, antifungal susceptibilities of the twelve A. fumigatus isolates and cyp51A gene analysis”. What is CLSI? Change: “cyp51A substitution”

- Figure 3. What's in the top left corner of the figure?

5) Discussion section:

- What did you mean: Importantly, these amino acid changes are not associated with those of the antifungal access receptors”.

- From the discussion of the results and conclusion it is not clear the significance of the results obtained. What practical significance do the results have? For example: “Among the 415 evaluated A. fumigatus isolates, only 2 carried the TR34/L98H/S297T/F495I mutation”. Or: “Moreover, this study reports the identification of the first environmental strain of A. fumigatus carrying the TR46/F495I mutation. The discovery holds significant implications for clinical management, especially if infections with this mutation pattern are reported”. What does this mean? What does this affect?

6) The number of references is too large for an experimental article.

Author Response

Dear Editorial Office,

We hereby resubmit the manuscript, ID jof-2815708, entitled “Uncovering a Novel cyp51A Mutation and Antifungal Resistance in Aspergillus fumigatus through Culture Collection Screening”, by Laís Pontes, Teppei Arai, Caio Augusto Gualtieri Beraquet, Ana Luisa Perini Leme Giordano, Franqueline Reichert-Lima, Edson Aparecido da Luz, Camila Fernanda de Sá, Larissa Ortolan Levy, Cibele Aparecida Tararam, Akira Watanabe, Maria Luiza Moretti, and myself, for further consideration.

We begin by thanking the Editorial Office and reviewers for their time and honest evaluation of the original version of this manuscript. The comments and suggestions have greatly improved the clarity and quality of our work. Therefore, we believe that the revised version meets the standards of the Journal of Fungi.

Dear Editorial Office,

We hereby resubmit the manuscript, ID jof-2815708, entitled “Uncovering a Novel cyp51A Mutation and Antifungal Resistance in Aspergillus fumigatus through Culture Collection Screening”, by Laís Pontes, Teppei Arai, Caio Augusto Gualtieri Beraquet, Ana Luisa Perini Leme Giordano, Franqueline Reichert-Lima, Edson Aparecido da Luz, Camila Fernanda de Sá, Larissa Ortolan Levy, Cibele Aparecida Tararam, Akira Watanabe, Maria Luiza Moretti, and myself, for further consideration.

Reviewer 3

The manuscript of Laís Pontes and co-authors is mainly a study of a collection of A. fumigatus strains. The authors analyzed the number of resistant strains and strains carrying mutations in the cyp51A gene. But it is still unclear from the work what fundamentally new data and conclusions were obtained in this work. The spread of azole-resistant fungal strains is widely known. And it is known that there are different mechanisms for the development of resistance. It is not clear why the authors chose to study mutations in cyp51A gene. The authors themselves write that mutations in this gene are not so common in their collection. And what conclusions can be drawn from the work done.

Authors: We thank Reviewer 3 for the thoughtful and constructive feedback on our manuscript. We appreciate the opportunity to address your comments. Our study aimed to contribute insights into the resistance landscape of Aspergillus fumigatus, focusing on environmental and clinical strains in Latin America, with a specific emphasis on Brazil. While azole-resistant strains are widely recognized, there is a gap in the literature regarding their prevalence and genetic characteristics in this specific geographical region.

We acknowledge that mutations in the cyp51A gene are not as common in our institution; however, we believe that reporting these findings is crucial. The rarity of such mutations in our collection emphasizes the need for continued surveillance, as their emergence could have significant implications for antifungal therapy.

The primary conclusions of our work stem from the identification of high MIC values in clinical strains, showcasing an increase compared to earlier publications, along with the discovery of two mutations in clinical strains. Additionally, the identification of an environmental mutation, previously unreported in the literature, adds a novel dimension to our understanding of A.fumigatus resistance patterns.

We hope these insights clarify the contributions of our study and its importance in the context of the limited data available for this region. 

The main comments

- The title of the manuscript is too long.

Authors: Thank you for the feedback regarding the title length, we have revised the title to address this concern. The updated title is now "Uncovering a Novel cyp51A Mutation and Antifungal Resistance in Aspergillus fumigatus through Culture Collection Screening”. 

- Abstract.

-It is not clear from the abstract what cyp51A gene is and what it is responsible for. About mutation: TR46/F495I requires clarification (tandem repeat of 46 bp?).

Authors: We have included information about the cyp51A gene in the abstract for better clarity. Regarding the TR46/F495I mutation, we acknowledge your point. Due to character limitations in the abstract, we could not provide a detailed explanation. However, TR46 refers to a tandem repeat of 46 base pairs in the cyp51A gene, resulting in an F495I amino acid change. This terminology is consistent with other literature where various authors refer to such mutations using similar nomenclature. We hope this additional information addresses your concerns, and we appreciate your understanding of the constraints posed by abstract length limitations.

-It is not clear from the conclusion what new has been done in this work. The spread of resistant strains of fungi is widely known.

Authors: We have revised the conclusion of the abstract to provide a clearer outline of the novel contributions made by our work. While it is true that the spread of resistant fungal strains is recognized globally, our study is important for Latin America, especially Brazil. In this region, only two published papers on resistance in Aspergillus fumigatus exist, and both were authored by our research team. Our findings, including the identification of a new type of mutation TR46/F495I (originally isolated in 1998), add important information for mycologists and clinicians. It is essential to note that the first Aspergillus fumigatus strain resistant to azoles with mutations in the cyp51A gene was described by a Dutch research group in the late 90s. Therefore, our discoveries represent a significant contribution to the field and build upon this important early work. We hope this clarification emphasizes the importance of our study in the context of the limited data available in Latin America.

3) Materials and methods section.

- It’s still not entirely clear what MEC is and what it shows.

Authors: Thank you for bringing this to our attention. We have revised the section to enhance clarity. MEC, which stands for Minimum Effective Concentration, indicates the concentration at which a noticeable change in the growth pattern of fungi occurs. This effect was observed specifically in response to echinocandins, with Micafungin and Caspofungin being the only tested representatives in this manuscript. We hope this clarification better conveys the significance and interpretation of MEC in our study.

- “Quality controls, including Aspergillus flavus ATCC 204304, Candida parapsilosis ATCC 167 22019, and Candida krusei ATCC 6258 were incorporated in each test.” Why was not the collection strain Aspergillus fumigatus used for this purpose?

Authors: In our study, we followed the guidance provided in the CLSI M38-A3 document, which recommends Aspergillus flavus ATCC, Candida parapsilosis ATCC, and Candida krusei ATCC as quality controls for antifungal susceptibility testing. While we did not include Aspergillus fumigatus, we diligently adhered to the guidelines, utilizing the strains recommended for quality control purposes. This approach ensures the reliability and consistency of our antifungal susceptibility testing procedures in line with established standards. 

4) Results section.

- There is no description of the actual results obtained. Some information should be transferred from the Discussion section or the Results and Discussion sections should be combined.

Authors: Thank you for your comment. We believe our results are adequately presented and described in the tables and relevant sections of the manuscript. If there are specific details or aspects you find unclear, we would greatly appreciate more specific guidance so that we can address your concerns effectively. Additionally, we are open to suggestions on how to enhance the presentation of our results, and we will consider the possibility of combining the Results and Discussion sections for better coherence. 

- Figure 2. Replace A1, A2, B1, B2 with A, B, C, D. What exactly is on the Petri dishes? What do the numbers on the petri dishes correspond to? What is the difference between A1 and A2 (B1 and B2)?

Authors: We appreciate your feedback. In response to your suggestion, we have revised the figure and legend to replace A1, A2, B1, and B2 with A, B, C, and D for improved clarity. Additionally, the legend now provides explicit information about the content of the Petri dishes. We trust that these modifications address your concerns and enhance the clarity of the figure. 

- Table 1. Сheck the table name. What values ​​are given in the table – MICs or MECs? What is “MIC/MEC Distribution (μg/mL)”?

Authors: Thank you for your observation. We have revised the table name to reflect the distinction between MIC and MEC values. In our analysis, we considered different drug classes, including azoles, polyenes, and echinocandins. For azoles (Itraconazole, Voriconazole, and Posaconazole), and Amphotericin B, we employed Minimum Inhibitory Concentration (MIC), indicating the concentration where 100% inhibition of fungal growth occurs. Conversely, for echinocandins, we used Minimum Effective Concentration (MEC), representing the concentration causing a change in the growth pattern of fungi. To account for these two different types of measurements, we have followed your suggestion and updated the table name to "MIC/MEC Distribution (μg/mL)." We agree that this modification provides clarity on the values presented in the table and the specific antifungal classes. 

- There are no table S1,S2 legends.

- Сhange the name “3.4. cyp51A Analysis”. Аnalysis of what?

Authors: Thank you for bringing attention to these details. Following your suggestion, we have now added the legends for Tables S1 and S2 and provided information and context for readers regarding the cyp51A analysis. 

- Table 2. Сhange table name “In vitro, antifungal susceptibilities of the twelve A. fumigatus isolates and cyp51A gene analysis”. What is CLSI? Change: “cyp51A substitution”

Authors: Thank you for your feedback. In response to your suggestions, we have revised the name of Table 2 to "Antifungal susceptibilities of resistant A. fumigatus isolates and cyp51A gene analysis" as per your recommendation. We trust that these adjustments enhance clarity, and we appreciate your attention to these details. 

Regarding your inquiry about CLSI, it stands for the Clinical and Laboratory Standards Institute. It is an organization responsible for developing global standards for clinical and laboratory testing across various healthcare domains. CLSI provides comprehensive guidelines and standards to ensure the quality and consistency of laboratory testing. These standards and guidelines are designed to assist laboratories in conducting tests in a standardized and reproducible manner, thereby contributing to the reliability of test results.

- Figure 3. What's in the top left corner of the figure?

Authors: We removed the item that was in the left corner of the figure.

5) Discussion section:

- What did you mean: “Importantly, these amino acid changes are not associated with those of the antifungal access receptors”.

Authors: This statement implies that, despite the presence of specific point mutations in the cyp51A gene, these specific mutations do not alter any azole antifungal receptors. In other words, even though these mutations occur in the cyp51A gene, they do not result in significant changes to the receptors targeted by azole antifungal drugs. Despite the mutation being in the cyp51A gene, the antifungal can still effectively exert its action because there is no relevant alteration in the drug's main target within the receptors.

- From the discussion of the results and conclusion it is not clear the significance of the results obtained. What practical significance do the results have? For example: “Among the 415 evaluated A. fumigatus isolates, only 2 carried the TR34/L98H/S297T/F495I mutation”. Or: “Moreover, this study reports the identification of the first environmental strain of A. fumigatus carrying the TR46/F495I mutation. The discovery holds significant implications for clinical management, especially if infections with this mutation pattern are reported”. What does this mean? What does this affect?

Authors: We believe that the discussion and conclusion sections provide clarity regarding the significance of the results obtained. The identification of two A. fumigatus isolates carrying the TR34/L98H/S297T/F495I mutations in the cyp51A gene is very significant. These two strains were associated with an infection in a patient who came to obit due to resistance to azoles.

Additionally, the discovery of the TR46/F495I environmental mutation in a strain preserved since 1998 represents another significant finding. While the number of strains with mutations and resistance in this manuscript is limited, it is crucial to emphasize the importance of resistance surveillance for A. fumigatus. This surveillance is essential globally, as highlighted by the World Health Organization (WHO) in a publication last year, due to the threat it poses to public health.

Concerning the highlighted phrase "The discovery holds substantial implications for clinical management, especially if infections with this mutation pattern are reported.", this is critical because understanding the susceptibility pattern of strains, as detailed in this work, is essential for informing and optimizing the clinical management of patients. It is relevant when infections with specific mutation patterns, such as TR46/F495I, are reported.

We hope this clarification provides a more objective overview of the practical significance of our results, and we appreciate your attention to these details.

6) The number of references is too large for an experimental article.

Authors: Thank you for your feedback. We acknowledge the comment about the number of references. However, it's important to note that all references included in this manuscript are essential for providing a comprehensive background in the introduction, detailing the methods in the materials section, and supporting the discussion. Each reference contributes significantly to the contextualization, methodology, and interpretation of the experimental findings. We believe that the inclusion of these references enhances the robustness and depth of our work. If there are specific concerns or if a more concise approach is recommended, we are open to further discussion and adjustments. 

The revised manuscript can be found in the attachment.

Round 2

Reviewer 2 Report

Comments and Suggestions for Authors

I have no further comments

Author Response

We appreciate your review.

Reviewer 3 Report

Comments and Suggestions for Authors

The authors have made a number of corrections, but the manuscript still needs to be improved.

Abstract section

“Through resistance surveillance in our institution's culture collection, it was possible to identify a new set of TR46/F495I mutations in an isolate recovered from the hospital environment that had been stored since 1998, and TR34/L98H/S297T/F495I were also identified in isolates clinicians.”

Rephrase, the meaning of the sentence is unclear.

Results section

There are no references in the text to Tables 2 and 3.

About resistant strains in Table 3. Authors write “Breakingpoint for Resistance = >2µg/mL for ITC and VRC; POS >0.5µg/mL; AMB >1µg/mL”. But according to the data in Table S1, not all resistant strains are included in the Table 3. Why? Сlarify the principle of data inclusion.

About Table 3 additionally. According to Table S1 resistant strain 3760 does not contain a mutation in cyp51A gene. At the same time sensitive strain 3768 contains this mutation. Clarify this information.

Discussion section

What did you mean: “Importantly, these amino acid changes are not associated with those of the antifungal access receptors”.

Authors: This statement implies that, despite the presence of specific point mutations in the cyp51A gene, these specific mutations do not alter any azole antifungal receptors. In other words, even though these mutations occur in the cyp51A gene, they do not result in significant changes to the receptors targeted by azole antifungal drugs. Despite the mutation being in the cyp51A gene, the antifungal can still effectively exert its action because there is no relevant alteration in the drug's main target within the receptors.

The cyp51A gene encodes an enzyme. And azoles inhibit the enzymatic activity of lanosterol 14-α-demethylase. Therefore, the word receptor must be replaced by “target” or “enzyme”.

Conclusion

The conclusion of the manuscript should be rewritten because it does not reflect the significance of the findings. For example, the sentence “Among the 415 evaluated A. fumigatus isolates, only 2 carried the TR34/L98H/S297T/F495I mutation.” does not reflect the significance of the results obtained.

Additionally, it is known that resistance to fungi develops through different mechanisms, for example, not only mutations in cyp51A gene, but also an increase in its expression or an increase in the expression of transmembrane transporter genes take place. The authors examined only mutations in cyp51A gene and found that prevalence of such mutations in the collection of strains was low. Therefore, in the absence of all data, it cannot be concluded that increase in MIC values is associated specifically with mutations in cyp51A gene. The collection contains resistant strains (five stains in Table2 and, for example, stains 2807 and 2814 from Table S1) that do not carry mutations in cyp51A gene. In addition, sensitive strains (for example, stains 469, 641, 708 from TableS1) with mutations in this gene are present in collection. In my opinion, in conclusion it is worth making other accents.

Author Response

The authors have made a number of corrections, but the manuscript still needs to be improved.

Authors: We appreciate your time and effort in reviewing our manuscript and providing feedback. We are grateful for your insightful comments, which have contributed to enhancing the quality of our work. We have carefully considered each of your suggestions and have implemented the recommended corrections to the best of our ability. Your feedback has been invaluable in refining the content and addressing the identified issues. We believe that the revised manuscript now meets the standards set by the Journal of Fungi.

Abstract section

“Through resistance surveillance in our institution's culture collection, it was possible to identify a new set of TR46/F495I mutations in an isolate recovered from the hospital environment that had been stored since 1998, and TR34/L98H/S297T/F495I were also identified in isolates clinicians.”

Rephrase, the meaning of the sentence is unclear.

 Authors: Thank you for your feedback. We have carefully reviewed the sentence and, to enhance clarity, we have revised it as follows:

"By conducting resistance surveillance within our institution's culture collection, we successfully identified a novel TR46/F495I mutation in an isolate retrieved from the hospital environment, which had been preserved since 1998. Moreover, clinical isolates were found to exhibit TR34/L98H/S297T/F495I mutations".

Results section

There are no references in the text to Tables 2 and 3.

 Authors: Thank you for bringing to our attention the absence of references in the text to Tables 2 and 3. Following your observation, we have addressed this concern.

About resistant strains in Table 3. Authors write “Breaking point for Resistance = >2µg/mL for ITC and VRC; POS >0.5µg/mL; AMB >1µg/mL”. But according to the data in Table S1, not all resistant strains are included in the Table 3. Why? Clarify the principle of data inclusion.

Authors: Thank you very much for your review. In the manuscript we do not have Table 3, if you refer to Figure 3 or Table 2, in both titles the information included that only the cyp51A gene of isolates that demonstrated resistance to azoles was analyzed. In Table S1 there are isolates resistant to Amphotericin B. For them, the cyp51A gene was not analyzed because resistance to amphotericin is not linked to this gene.

About Table 3 additionally. According to Table S1 resistant strain 3760 does not contain a mutation in cyp51A gene. At the same time sensitive strain 3768 contains this mutation. Clarify this information.

Authors: Upon careful examination of your comment, it appears there might be a misunderstanding. We want to clarify that our manuscript does not include Table 3. If your reference pertains to either Figure 3 or Table 2, we wish to emphasize that the title of both explicitly states that the analysis is focused on the cyp51A gene of isolates demonstrating resistance to azoles. Furthermore, regarding the resistant strains not included in Table 3 but present in Table S1, we would like to explain that resistance to Amphotericin B is not directly associated with the cyp51A gene. Additionally, we carefully examined the supplementary material and observed that certain data points had changed their placement within the tables. As a result, we conducted a meticulous review and made necessary corrections to ensure accuracy and consistency in Table S1. We appreciate your attention to detail and are grateful for the opportunity to address this matter. We appreciate your attention to detail and are grateful for the opportunity to address this matter.

Discussion section

What did you mean: “Importantly, these amino acid changes are not associated with those of the antifungal access receptors”.

Authors: This statement implies that, despite the presence of specific point mutations in the cyp51A gene, these specific mutations do not alter any azole antifungal receptors. In other words, even though these mutations occur in the cyp51A gene, they do not result in significant changes to the receptors targeted by azole antifungal drugs. Despite the mutation being in the cyp51A gene, the antifungal can still effectively exert its action because there is no relevant alteration in the drug's main target within the receptors.

The cyp51A gene encodes an enzyme. And azoles inhibit the enzymatic activity of lanosterol 14-α-demethylase. Therefore, the word receptor must be replaced by “target” or “enzyme”.

 Authors: Done.

Conclusion

The conclusion of the manuscript should be rewritten because it does not reflect the significance of the findings. For example, the sentence “Among the 415 evaluated A. fumigatus isolates, only 2 carried the TR34/L98H/S297T/F495I mutation.” does not reflect the significance of the results obtained.

Authors: Thank you for your evaluation and constructive feedback. In response to your suggestion, we have carefully reconsidered and revised the conclusion paragraph of our manuscript. The updated conclusion now reads as follows:

"This study investigated the epidemiological profile of A. fumigatus clinical isolates in a Brazilian hospital, covering two time periods (1998-2005 and 2014-2021). Among the 415 isolates analyzed, two exhibited the TR34/L98H/S297T/F495I mutation in the cyp51A gene, and notably, this study identified the first environmental strain of A. fumigatus carrying the TR46/F495I mutation within the hospital. The presence of this mutation pattern has significant implications for clinical management, especially if infections with such mutations are reported. Despite a low prevalence of cyp51A mutations in our center, the observed high MIC values for azoles may be attributed to other resistance mechanisms, such as efflux pumps, mutations in the cyp51B gene, or hmg1 gene. The identification of elevated MIC values for both AMB and azoles suggests a changing resistance pattern in our institution, emphasizing the urgency of discovering new antifungal drugs and resistance mechanisms. Continued surveillance of new resistant isolates is deemed necessary.”

We hope that this revised conclusion now more accurately reflects the significance of our findings. We sincerely appreciate your dedication to enhancing the quality of our manuscript.

Additionally, it is known that resistance to fungi develops through different mechanisms, for example, not only mutations in cyp51A gene, but also an increase in its expression or an increase in the expression of transmembrane transporter genes take place. The authors examined only mutations in cyp51A gene and found that prevalence of such mutations in the collection of strains was low. Therefore, in the absence of all data, it cannot be concluded that increase in MIC values is associated specifically with mutations in cyp51A gene. The collection contains resistant strains (five stains in Table2 and, for example, stains 2807 and 2814 from Table S1) that do not carry mutations in cyp51A gene. In addition, sensitive strains (for example, stains 469, 641, 708 from TableS1) with mutations in this gene are present in collection. In my opinion, in conclusion it is worth making other accents.

Authors: We sincerely appreciate your thorough review and the insightful comments provided, especially concerning the conclusion of our manuscript. We value your expertise and have carefully considered your suggestions. In response to your concerns, we have made significant revisions to the conclusion and discussion sections of the manuscript. We now explicitly acknowledge in the text that resistance to azoles may develop through various mechanisms, as described in the existing literature.

Our study primarily focused on the analysis of the cyp51A gene, given its common association with azole resistance mutations. However, we now recognize and highlight the complexity of resistance development, encompassing factors such as an increase in the expression of the cyp51A gene and the overexpression of transmembrane transporter genes.

Moreover, we have thoroughly reviewed Table S1, addressing and correcting all discrepancies to ensure the accurate presentation of the data.

We trust that these refinements address your concerns and enhance the clarity and accuracy of our manuscript. Your feedback has been invaluable, and we sincerely appreciate your commitment to improving the quality of our work. Thank you once again for your review.